# Dietary Microencapsulated Blend of Organic Acids and Plant Essential Oils Affects Intestinal Morphology and Microbiome of Rainbow Trout (*Oncorhynchus mykiss*)

**DOI:** 10.3390/microorganisms9102063

**Published:** 2021-09-30

**Authors:** David Huyben, Marcia Chiasson, John S. Lumsden, Phuc H. Pham, Mohiuddin A. Kabir Chowdhury

**Affiliations:** 1Department of Animal Biosciences, University of Guelph, Guelph, ON N1G 2W1, Canada; 2Ontario Aquaculture Research Centre, Office of Research, University of Guelph, Elora, ON N0B 1S0, Canada; marciach@uoguelph.ca; 3Department of Pathobiology, University of Guelph, Guelph, ON N1G 2W1, Canada; jsl@ovc.uoguelph.ca (J.S.L.); phpham@uoguelph.ca (P.H.P.); 4Jefo Nutrition Inc., Saint-Hyacinthe, QC J2S 7B6, Canada; kchowdhury@jefo.ca

**Keywords:** 16S rDNA sequencing, aquaculture, feed additive, gut inflammation, intestinal health, viable plate counts

## Abstract

A study was conducted on 500 juvenile rainbow trout (122 ± 4 g) fed either a control diet or a treatment diet containing 300 mg/kg of a microencapsulated blend of organic acids and essential oils to elucidate effects on intestinal morphology and microbiome. Proximal intestinal villi length was significantly increased in fish fed the treatment diet. Despite no differences in gut inflammation scores, edema, lamina propria inflammation and apoptosis were completely absent in the distal intestine of fish fed the treatment diet. Next-generation sequencing of the 16S rDNA showed no differences in alpha and beta diversity, and gut bacteria were mainly composed of Firmicutes, Bacteroidetes and Proteobacteria. On the genus level, LefSe analysis of indicator OTUs showed *Bacteroides*, *Sporosarcina*, *Veillonella*, *Aeromonas* and *Acinetobacter* were associated with the control diet whereas *Streptococcus*, *Fusobacterium* and *Escherichia* were associated with the treatment diet. *Aeromonas hydrophila* and *Acinetobacter* spp. are opportunistic pathogens and several strains have been found to be resistant to antibiotics. The increase in villi length and reduction of specific pathogens indicates that feeding a microencapsulated blend of organic acids and essential oils improves gut health and may serve as a part of an effective strategy to reduce antibiotic use in aquaculture.

## 1. Introduction

Many animal and plant proteins have been used to reduce the use of fish meal in aquafeed. However, the use is severely restricted because of their imbalanced amino acid profile, presence of anti-nutrients, poor palatability and reduced digestibility [1]. Several nutritional strategies such as supplementation of amino acids and enzymes have been adopted to improve the utilization of these protein sources and to improve nutrient uptake [2,3]. In addition to nutritional concerns, alternative strategies are needed to reduce antibiotic use and avoid the transfer of antibiotic resistant genes in aquaculture. Recently, the addition of organic acids and their salts to aquafeed as well as phytogenic compounds or essential oils have been evaluated to improve gut health and microbial communities in aquatic species [4,5,6,7]. Organic acids and essential oils hold high potential to improve nutrient uptake and gut health of fish as alternative to antibiotics, although most studies to date have focused on warm water fish and shrimp species.

Organic acids are composed of short-chain compounds (C1–C7), such as benzoic, formic, lactic and propionic acids, which have traditionally been used as storage preservatives in food and feed [8]. The use of organic acids in aqua-feeds has been recently reviewed [9] and many studies have reported that organic acids can significantly enhance the intestinal health of both cold- and warm-water fish species such as rainbow trout, *Oncorhynchus mykiss* [4], rohu (*Labeo rohita*) [10,11], and red sea bream (*Pagrus major*) [12,13]. Contradictory results have also been reported, which seems to depend on the aquatic animal species, type, composition, and concentration of organic acids and the culture conditions used [14].

Essential oils are natural complex mixtures of volatile, lipophilic, odoriferous and liquid substances obtained from plant raw materials [15]. More than 3000 distinct compounds have been detected in essential oils classified as terpene hydrocarbons, simple and terpene alcohols, aldehydes, ketones, phenols, esters, ethers, oxides, peroxides, furans, lactones, coumarins, and sulfur compounds [16]. Essential oils in aquafeed have been recently reviewed [17,18,19]. Feeding essential oils have been found to improve disease resistance and modulate the gut microbiome of rainbow trout [7,20,21,22]. However, research on the combined use of essential oils and organic acids is lacking.

In the past decade, researchers have been able to map and investigate dietary changes on the microbes in the gut (referred to as the gut microbiome) using next-generation sequencing [23,24]. The microbes in the gut and their metabolites have enormous effects on health status of the host via modulation of the immune system, production of nutrients and other metabolic functions [25]. In general, essential oils have been shown to be more antagonistic towards pathogens than commensal bacteria, inhibiting certain bacterial groups in the gut while probiotic microbes can proliferate [26,27]. The cell wall structure of Gram-positive bacteria allows hydrophobic molecules, such as essential oils, to easily penetrate and act both on the cell wall and in the cytoplasm [28]. In addition, stimulating effects of essential oils allow the microbiota to modulate and improve digestion and absorption of nutrients, which may supply more amino acids and fatty acids for protein and lipid syntheses [29]. However, studies on rainbow trout and red drum (*Sciaenops ocellatus*) have not found significant effects of essential oils on the gut microbiome [30,31], although older gel electrophoresis methods were used.

Microencapsulation has become one of the most popular and practical approaches to deliver bioactive compounds to specific regions of the gastro-intestinal tract in farmed animals, e.g., swine [32,33]. A hydrogenated-fat based microencapsulation technique, as described in Chowdhury et al. [34], was used in this study to protect the blend of organic acids and essential oils as well as release it into the intestine.

The objective of this study was to determine the effect of a microencapsulated blend of organic acid and essential oils on the intestinal histology and microbiome of rainbow trout fed a high plant protein diet. Histological morphometrics and scoring as well as next-generation sequencing of the 16S rDNA and viable plate counts were used to analyze intestinal samples after fish were fed the control and experimental diets.

## 2. Materials and Methods

### 2.1. Fish Facilities

Rainbow trout were hatched at the Ontario Aquaculture Research Centre (Alma, ON, Canada) and 500 juvenile, mixed sex fish (122 ± 4 g) were randomly distributed across ten 1-m tanks that contained 330 L of water. Fish were acclimated for 14 days in the experimental tanks and fed a high plant protein commercial diet (Table 1) containing 40.5% plant protein ingredients (soybean meal–25.5%, corn DDGS–10%, and corn gluten meal–5%) using automatic belt feeders. Water was analysed weekly from the flow-through freshwater system that had a mean (±SD) temperature of 8.5 ± 0.1 °C, dissolved oxygen of 9.5 ± 0.3 mg/L, pH of 8.0 ± 0.1, total suspended solids of 0.4 ± 0.2 mg/L and flow rate of 11.2 ± 0.2 L/min. The experimental photoperiod was 12:12 light-dark cycle utilizing LED lights with a 60 min ramp time to simulate dawn and dusk. The animal experiment was reviewed and approved by the Animal Care Committee at the University of Guelph under Animal Utilization Protocol #4503.

### 2.2. Diets and Feeding

The same commercial diet was top coated with 300 mg/kg (0.03% of the diet) of microencapsulated proprietary blend of four organic acids (sorbic acid, fumaric acid, malic acid, and citric acid) and three essential oils (thymol, vanillin, and eugenol) by heating up at 60 °C for 2 min with 50 mL of fish oil and top coating the feed. A control diet was mixed with the same amount of fish oil without the additive. The two diets were randomly assigned to the fish tanks with five (*n* = 5) replicates each. Fish were hand-fed to satiety twice weekly, rations were altered, and belt feeders were used to feed a 95% ration for five days per week. The belt feeders delivered two meals daily at approximately 9:00 and 14:00 h for 28 days.

### 2.3. Sample Collection

At the end of the experiment, each fish was measured for weight and total length. Six fish per tank were euthanized with MS-222, the gill arches were cut, and the abdomen was dissected. From two fish per tank (*n* = 10), 1-cm sections of the proximal and distal intestines were collected from two fish per tank, cut open longitudinally and stored in 10% formalin. From an additional two fish per tank (*n* = 10), the distal intestine was cut with a sterile scalpel 0.5 cm before the anus and faeces (or digesta) were collected by squeezing the contents into a sterile tube, which was placed on ice. All samples were submitted to the Animal Health Lab at the University of Guelph (Guelph, ON, Canada) and stored at room temperature (histology), 4 °C (cultures) or −80 °C (16S). Viscera and liver were weighed from two fish per tank (*n* = 10) used to calculate viscerosomatic index (VSI) and hepatosomatic index (HSI):VSI = (viscera weight/fish weight) × 100
HSI = (liver weight/fish weight) × 100

### 2.4. Histology of Proximal and Distal Intestine

Intestinal samples were cut perpendicular to approximately 2–3 mm in four separate sections. Samples were serially dehydrated in 70–100% ethanol and embedded in paraffin using a Sakura VIP 6 tissue processor. Sectioning was performed using an automated microtome Leica RM 2255 at 4-µm and hematoxylin and eosin staining was performed using the Leica ST 5020 Multistainer and Leica CV5030 Coverslipper. Slides were observed under the Olympus BX45 light microscope using the 4x objective. Pictures were taken using the Olympus DP71 camera and Olympus cellSens software. The polyline tool in the software was used to measure simple folds (villi), but not complex folds. The length of folds was measured from the stratum compactum, following the lamina propria to the fold terminus. The width of folds was measured as close to the midway from stratum compactum to villi terminus whenever possible, or in cases where the midway does not show the proper villi width size (due to overlap of multiple villi or other issues), measurement was taken at the next best location closest to the midway. There were at least four sections for each sample/slide and measurements were taken from sections until 10 measurements were performed.

Numerous slides were screened randomly for lesions and normal features that might vary between the material examined. Five full-length simple intestinal folds (not complex folds) were scanned for the criteria below for one of the four sections per slide. The most complete tissue section with as few artefacts as possible was chosen. The criteria and scales that were used were:

All histological scoring was based on a scale from 0 to 3. Using a 4× objective lens, edema and inflammation of lamina propria, submucosa and serosa/musculature were scored based on 0 = none, 1 = mild, 2 = moderate, and 3 = severe or more extensive than 2. Using a 10x objective lens, epithelial vacuolization was scored based on 0 = few vacuoles filled, 1 ≤ 1/3 filled, 2 ≤ 1/2 filled, 3 ≥ 1/2 filled and folded epithelial. Goblet cells were scored in a similar way. Using a 20× objective lens, the numbers of mitotic figures were counted for the five intestinal folds and averaged based on 0 ≤ 1, 1 = 1–2 per fold, 2 = 2–3 per fold, and 3 ≥ 3 per fold. Epithelial necrosis or apoptosis was scored in a similar way.

### 2.5. Viable Plate Counts of Bacteria

Aseptically, 100 mg of feces were diluted with 900 µL of PBS and vortexed to create the first dilution, and then repeated two more times to make 1/100 and 1/1000 dilutions. For each dilution, 100 µL were pipetted and spread onto duplicate plates of TSA (Oxoid, Thermo Fisher, Nepean, ON, Canada) using a hockey stick and plates spinner. Plates were incubated at 22 °C for 24 h ± 3 h. Counts less than 25 or more than 250 colonies as viable plate counts (VPC) were excluded and the mean was divided by the dilution factor (https://www.fda.gov/food/laboratory-methods-food/bam-chapter-3-aerobic-plate-count accessed on 13 May 2021). At random, five colonies were chosen and directly spotted on stainless steel targets and analysed using a MALDI-TOF with BMT Compass software (Bruker, Billerica, MA, USA).

### 2.6. Extraction and Sequencing of 16S rDNA Bacteria

Approximately 200 mg of faeces was extracted using a QIAamp Fast DNA Stool Mini kit (Qiagen Inc, Toronto, ON, Canada) according to the manufacturer’s instructions. The DNA concentration was quantified using a Quibit 2.0 fluorimeter (Invitrogen, Thermo Fisher Scientific, Waltham, MA, USA). A two-stage PCR was performed to target the V3-V4 region of the 16S rDNA according to the 16S Library Preparation Guide (Illumina Inc, San Diego, CA, USA). In brief, 25 µL reactions consisting of 2.5 µL template DNA (5 ng/μL in 10 mM

Tris pH 8.5, 5 µL (1 μM) of each forward primer (341F; TCGTCGGCAGCGTCAGATGTGTATAAGAGACAGCCTACGGGNGGCWGCAG) and reverse primer (785R; GTCTCGTGGGCTCGGAGATGTGTATAAGAGACAGGACTACHVGGGTATCTAATCC) [35] and 12.5 µL of 2× KAPA HiFi HotStart ReadyMix (Sigma-Aldrich, Oakville, ON, Canada). The PCR conditions were 95 °C for 3 min followed by 40 cycles of 95 °C for 30 s, 55 °C for 30 s and 72 °C for 30 s with a final step of 72 °C for 5 min. Amplicons were confirmed on a 1% agarose gel alongside negative controls of nuclease free water (NTC) and no enzyme control (NEC). Positive controls of feed were included as well. Samples were purified with Agencourt AMPure XP beads and 80% ethanol (Beckman Coulter, Indianapolis, IN, USA), according to the manufacturer’s instructions. A second PCR was performed with the above conditions, except only for 8 cycles and the forward and reverse primers consisted of different combinations of eight basepairs (bp) from the Nextera XT DNA Library Preparation kit (Illumina Inc) to individually index each sample. Samples were purified again as described above and quantified with a Quibit 2.0 fluorimeter (Thermo Fisher Scientific). Samples were diluted (normalized) to 4 nM and 10% phix control was spiked in to loading concentration of 6 pM. The library was sequenced on the Illumina MiSeq platform at the University of Guelph (Guelph, ON, Canada) to produce 2 × 300 bp pair-end reads with a MiSeq Reagent kit v3 of 600 cycles (Illumina Inc). Quality of sequence reads was examined using MultiQC (https://multiqc.info/ accessed on 23 May 2021).

### 2.7. Bioinformatics of 16S rDNA Bacteria

The 16S rDNA sequences were analysed using Mothur version 1.42.3 [36] according to the MiSeq SOP [https://www.mothur.org/wiki/MiSeq_SOP accessed on 29 May 2021] [37]. Sequence reads which were smaller than 300 bp, larger than 400 bp, had more than eight consecutive bp and were outside the V3-V4 region of the 16S rRNA were removed from the dataset. Filtered sequence reads were aligned to the SILVA reference database version 123 [38], pre-clustered to merge sequences with less than 2 bp difference and chimeras were removed using the open-source tool VSEARCH [39]. Sequences were classified using the RDP Bayesian Classifier trainset version 16.0 at a cut off of 80% [40] and taxon resembling chloroplasts, mitochondria, unknowns, archaea and eukaryotes were removed. All samples were normalized (subsampled) to the sample that had the lowest number of reads. Raw fastq files were deposited in the NCBI Sequence Read Archive (http://www.ncbi.nlm.nih.gov/bioproject/767341 accessed on 29 September 2021).

### 2.8. Statistical Analysis

Normal distribution and homogeneity of each dataset were determined using Shapiro–Wilk and Levene tests in RStudio version [41]. When needed, data were normalized by log or square-root transformation and all data are presented as means ± SE unless otherwise specified. A Student’s *T*-test was performed on normal data and a Wilcoxon test was performed on non-normal data. Alpha-diversity tables were created using Number of OTU’s, Good’s Coverage, Shannon and Choa-1 indices. Lefse and ANOSIM were used to analyse the indicator OTU and beta-diversity of the 16S rDNA dataset. *p*-values below 0.05 were considered significant.

## 3. Results

### 3.1. Gut Histology

Compared to the control diet, fish fed the treatment diet had higher villi length in both the proximal intestine (645 vs. 686 µm, *p* = 0.035) and distal intestine (989 vs. 1039 µm, *p* = 0.278). There was no difference (*p* > 0.05) between the villi width, edema, inflammation of serosa/submucosa/lamina, vacuolization, goblet cells, mitoses, and necrosis/apoptosis between the two diets (Table 2 and Figure 1). It was worth noting that edema, inflammation of the lamina propria and necrosis/apoptosis had a mean score of 0.0 in the distal intestine for fish fed the treatment diet.

### 3.2. Viable Plate Counts and MALDI-TOF of Gut Microbiome

Total viable counts of microbes in the fish gut ranged from 4.5–4.6 log CFU/mg, while no significant effect of diet was found (Figure 2). MALDI-TOF analysis of cultured isolates showed the gut mainly contained *Pseudomonas, Carnobacterium, Staphylococcus* and *Candida* (yeast) with a low abundance of *Bacillus, Brochothrix, Psychrobacter, Yarrowia* (yeast), *Acinetobacter* and *Aeromonas* (Figure 3). No significant differences in relative abundance of viable microbes were found (*p* > 0.168).

### 3.3. 16S rDNA Sequencing of Gut Microbiome

The 16S rDNA sequencing produced 2.9 million reads with a mean (±SD) of 207,012 ± 93,261 reads per sample. Chloroplast, mitochondria and eukaryotes were removed and reduced the amount of reads by 63.8% to a total of 1.0 million reads. All samples were normalized to the lowest sample that contained 25,530 reads per sample. The alpha diversity of bacteria in the faeces was not affected (*p* > 0.05) by diet (Table 3). On the phyla level, faeces were mainly composed of Firmicutes, Bacteroidetes and Proteobacteria with a lower abundance of Fusobacteria, Actinobacteria and Spirochaetes (Figure 4). On the genus level, faeces were mainly composed of *Bacteroides* and *Fusobacterium* with a lesser extent of *Streptococcus*, *Peptostreptococcus*, *Clostridium*, *Aeromonas* and *Lactobacillus* (Figure 5). There was no significant difference in beta-diversity using ANOSIM (*R* = 0.141, *p* = 0.156).

Analysis with LefSe noted several indicator bacteria species associated with each diet (Table 4). In the faeces, the treatment diet increased the abundance of *Streptococcus* and *Fusobacterium*, while the control diet had increased abundance of *Bacteroides*, *Sporosarcina*, *Veillonella*, *Aeromonas* and *Acinetobacter*.

### 3.4. Feeding and Body Indices

Final weight, final length, feed intake, body indices and survival were not significantly different (*p* > 0.05) between fish fed the control and treatment diets. For control and treatment groups, final weight was 178 and 174 ± 10 g (pooled SE), final fork length was 23.6 and 23.5 ± 0.5 cm, daily feed intake per fish was 1.86 and 1.88 ± 0.2 g, VSI was 11.5 and 11.4 ± 0.8, VSI was 1.5 and 1.7 ± 0.2 and survival was 97.6 and 98.8 ± 1.8%, respectively.

## 4. Discussion

In recent years, the combined use of hydrophobic essential oils with lipophilic organic acids in diets has received much attention for the potential synergistic and additive benefits on intestinal health in pigs, poultry and aquatic animals compared with individual essential oils and organic acids. The aim of the present study was to investigate effects of a proprietary blend of four organic acids (sorbic acid, fumaric acid, malic acid, and citric acid) and three essential oils (thymol, vanillin, and eugenol) on the gut health of rainbow trout, specifically influences on gut histology and the microbiome.

### 4.1. Gut Histology

Intestinal morphology, including villi height, is an important indicator of intestinal health, recovery and functionality and plays an important role in nutrient digestion and absorption [42]. In this study, feeding the combination of organic acids and essential oils significantly increased villi length in the proximal intestine (Table 2 and Figure 1). Similarly, microencapsulated sodium butyrate increased the villi height in common carp [43]. Increased intestinal villi height has also been observed in Pacific white shrimp (*Litopenaeus vannamei*) [6] and Nile tilapia hybrids [5] fed diets supplemented with a microencapsulated blend of the salts of the same organic acids as used in the present study. In addition to villi length, there were no signs of edema, inflammation of lamina propria and necrosis in the distal intestine in fish fed the treatment diet compared to their minor appearances in fish fed the control diet in the present study (Table 2). This underscores the efficiency of the microencapsulated product, where the active ingredients were designed to be released in the distal intestine of the animal [44]. A comparison to fish with intestinal inflammation, due to poor diet or rearing conditions, may have shown a significant beneficial effect. For example, Pelusio et al. [45] fed organic acids and essential oils (citric acid, sorbic acid, thymol and vanillin) to rainbow trout reared at 23 °C, a chronic stressor, found that these compounds reduced inflammatory activity in the intestinal mucosa.

The increased intestinal villi length and a tendency for reduced inflammation found in the present study may indicate increased oxidative capacity and intestinal function [46]. Pelusio, Rossi, Parma, Volpe, Ciulli, Piva, D’Amico, Scicchitano, Candela, Gatta, Bonaldo and Grilli [45] fed a microencapsulated blend of organic acids and essential oils to rainbow trout and found that gene up-regulation involved a limited number of cytokines showing the absence of a substantial inflammation process able to compromise the functional activity of the intestine. Feeding 0.5% essential oils from clove basil improved of phagocytic activity of Nile tilapia (*Oreochromis niloticus*) [47]. Feeding 0.02–0.05% of essential oils from oregano, lemongrass and geranium increased lysozyme and catalase activities in channel catfish (*Ictalurus punctatus*) [48] and Nile tilapia [49] compared to the basal diet. Pacific white shrimp fed a similar blend of organic acids and essential oils resulted in reduced expression of pro-inflammatory immune genes, elevated expression of lysozyme and catalase genes, increased serum phenoloxidase and glutathione peroxidase activities and higher disease resistance [50].

### 4.2. Gut Microbiome

The gut microbiota constitutes a highly complex ecosystem that interacts with the host and profoundly affects the physiological, immunological, nutritional and metabolic status of the host [24]. Only a handful of studies have investigated dietary effects on the gut microbiome of rainbow trout using next-generation sequencing [51,52,53,54,55]. In agreement with these studies, the present study found high relative abundances of Firmicutes and Proteobacteria phyla, specifically genera *Fusobacterium*, *Streptococcus*, *Lactobacillus, Pseudomonas* and *Vagococcus* (Figure 4 and Figure 5). The comparison of gut microbes found using MALDI-TOF compared to next-generation sequencing (Figure 3 and Figure 5) clearly illustrates the bias of culture based methods that have been previously demonstrated [56]. The culture-based method did identify commonly found bacteria in the gut of rainbow trout in high abundance (2–14%), such as *Carnobacterium*, *Staphylococcus* and *Bacillus*, although these were less abundant when using the next-generation sequencing method (0.1–0.3%). In addition, next-generation sequencing found significant differences in indicator species (Table 4), whereas the culture-based method did not.

In vitro studies have found that essential oils, particularly thymol, can target cell membranes of pathogenic bacteria and cause significant damages in the membrane permeability, integrity and composition [57,58]. The cell wall structure of Gram-positive bacteria allows hydrophobic molecules, such as essential oils, to easily penetrate and act both on the cell wall and in the cytoplasm [28]. Disruption of cell membrane increases penetrability of organic acids and in their undissociated form, alters proton and associated anion concentration in the cytoplasm of microbes [59]. In addition to these anti-microbial effects, organic acids reduce digesta pH, increase pancreatic secretion, and trophic effects on the gastrointestinal mucosa. Microencapsulation used in this study allows organic acids and essential oils to be released in the hindgut in their undissociated forms improving their efficacy compared to free form of these acids.

In the present study, feeding a microencapsulated blend of organic acids and essential oil reduced the abundance of *Aeromonas* (Table 4 and Figure 5) bacteria in the gut of rainbow trout. Further analysis using the Greengenes database revealed the most common *Aeromonas* species was *A. hydrophila*, a cause of ill health including tail and skin rot, and fatal haemorrhagic septicaemias in several fish species [60]. Treatment is difficult since *Aeromonas* strains are known for their enhanced capacity to acquire and exchange antibiotic resistance genes in aquatic environments [61]. This agrees with a previous study that fed 0.02–0.04% of essential oils from lemongrass and geranium and found reduced viable plates counts of *Aeromonas* in the gut of Nile tilapia [49]. In addition to *Aeromonas*, the treatment diet reduced the abundance of *Acinetobacter* (Table 4 and Figure 5). This is in agreement with a previous study that fed a similar blend of organic acids and essential oils to Pacific white shrimp [50]. Over the last 30 years, particular *Acinetobacter* species have emerged as opportunistic pathogens associated with nosocomial infections in humans with several strains resistant to antibiotics [62]. Therefore, feeding organic acids and essential oils represents one component of a strategy to reduce the abundance of pathogens and the potential of infection, particularly where antimicrobial resistance is problematic.

Previous studies also found that feeding fish essential oils and/or organic acids does not cause large shifts in gut microbial diversity and composition (Table 3 and Figure 4). Using electrophoresis methods, studies on rainbow trout and red drum have not found significant effects of essential oils on the gut microbiome [30,31]. Similarly, Pelusio, Rossi, Parma, Volpe, Ciulli, Piva, D’Amico, Scicchitano, Candela, Gatta, Bonaldo and Grilli [45] found no significant effect of organic acids and essential oils on the alpha-diversity (Shannon and Chao1) of gut microbiota in rainbow trout using next-generation sequencing. There was a trend towards reduction of *Streptococcus* in fish feed higher inclusion diets, whereas the present study found higher abundance of *Streptococcus* (Table 4). However, further analysis classifying sequences to the Greengenes database to determine the exact species showed *Streptococcus agalactiae* were numerically lower (*p* = 0.428) in fish fed the treatment diet, while *Streptococcus minor* and other unclassified *Streptococcus* spp. found in higher abundance (*p* > 0.05). *Streptococcus agalactiae* is a common fish pathogen and a recent study found that feeding a blend of five organic acids (formic acid, lactic acid, malic acid, tartaric acid and citric acid) improved the resistance of red hybrid tilapia to the pathogen [63]. Soltani et al. [64] found that essential oils (from Shirazi thyme and rosemary) reduced growth of *Streptococcus iniae*, a common pathogen found in rainbow trout, although this pathogen was not found in the present study. In addition, essential oils have been shown to be more antagonistic towards pathogens rather than commensal bacteria, inhibiting certain bacterial groups in the gut while probiotic microbes can proliferate in the gut of humans and swine [26,27].

## 5. Conclusions

Feeding a microencapsulated blend of organic acids and essential oils significantly affected the histology and the abundance of specific bacterial pathogens in the intestine of rainbow trout. This research demonstrates the benefit of encapsulating a small dose of organic acids and essential oils to bypass the early GI tract for release into targeted regions of the intestine. Increased villi length and the absence of inflammation in the intestine suggest improved gut health that may lead to improved oxidative capacity and gut function. The reduction in *Aeromonas hydrophila* and *Acinetobacter* spp. indicates this treatment may also be effective at reducing opportunistic pathogens in the gut of rainbow trout and may serve as part of an effective strategy to reduce antibiotic use in aquaculture.

## Figures and Tables

**Figure 1 microorganisms-09-02063-f001:**
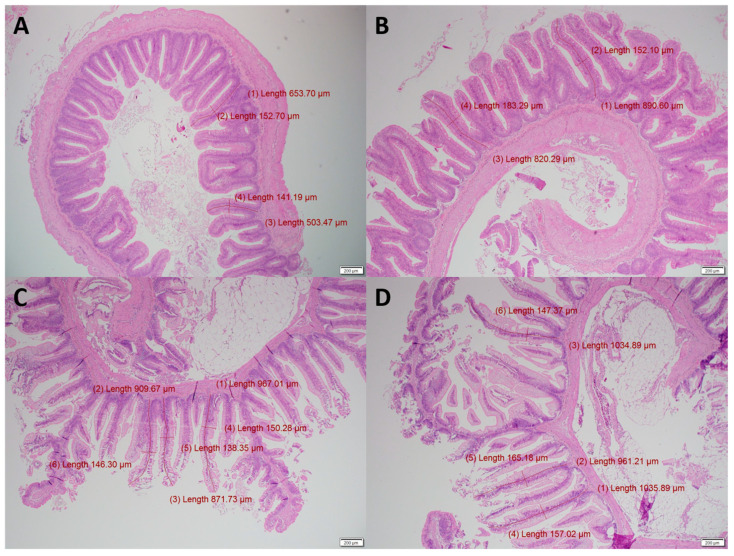
Histology slides of the proximal (**A**,**B**) and distal (**C**,**D**) intestine of rainbow trout fed the Control (**A**,**C**) and Treatment (**B**,**D**) diets. An example of inflammation of the lamina propria found in the Control group (left), but not the Treatment group (right). Villi length measurements are in red and were significantly longer (*p* < 0.05) in the proximal intestine of fish fed the treatment diet (**B**) compared to the control (**A**).

**Figure 2 microorganisms-09-02063-f002:**
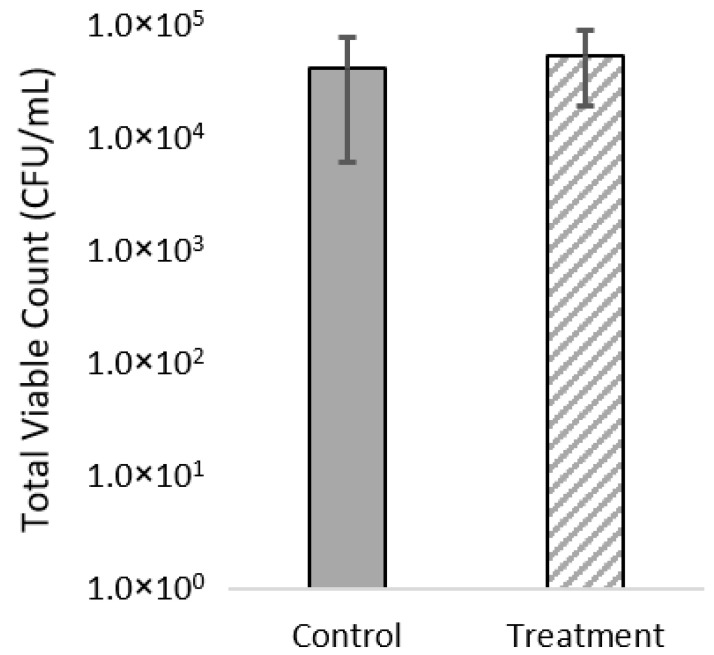
Counts of aerobic microbes from the gut digesta (*n* = 10) from rainbow trout cultured on TSA plates at 22 °C for 24 h (log-scale).

**Figure 3 microorganisms-09-02063-f003:**
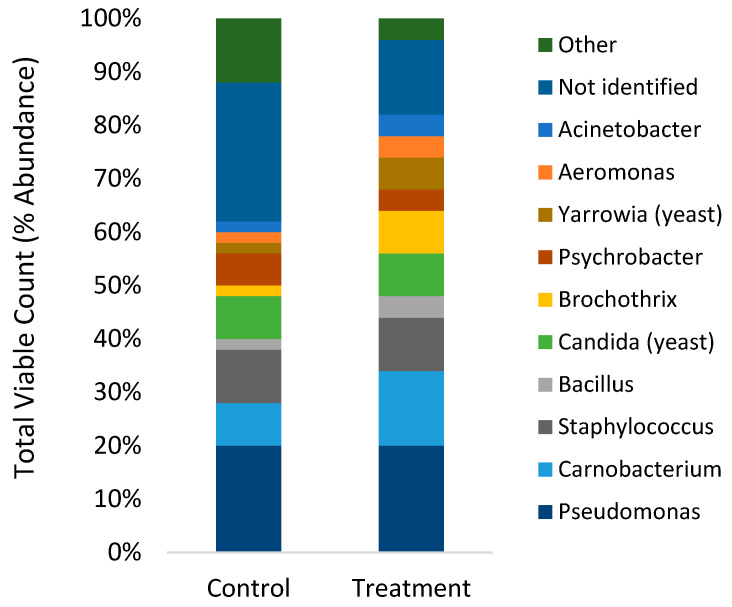
Relative abundance of viable microbes from the gut digesta (*n* = 10) from rainbow trout cultured on TSA plates and identified by MALDI-TOF.

**Figure 4 microorganisms-09-02063-f004:**
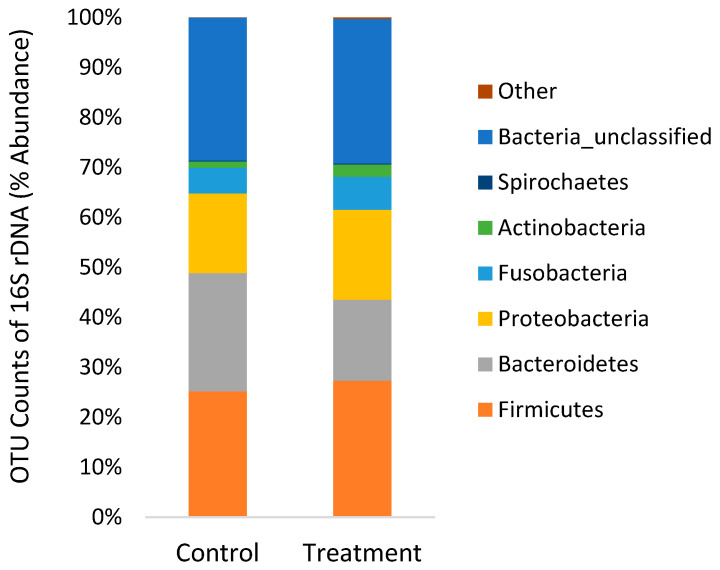
Relative abundance of OTUs on the phyla level clustered from 16S rDNA from the gut digesta (*n* = 5) of rainbow trout.

**Figure 5 microorganisms-09-02063-f005:**
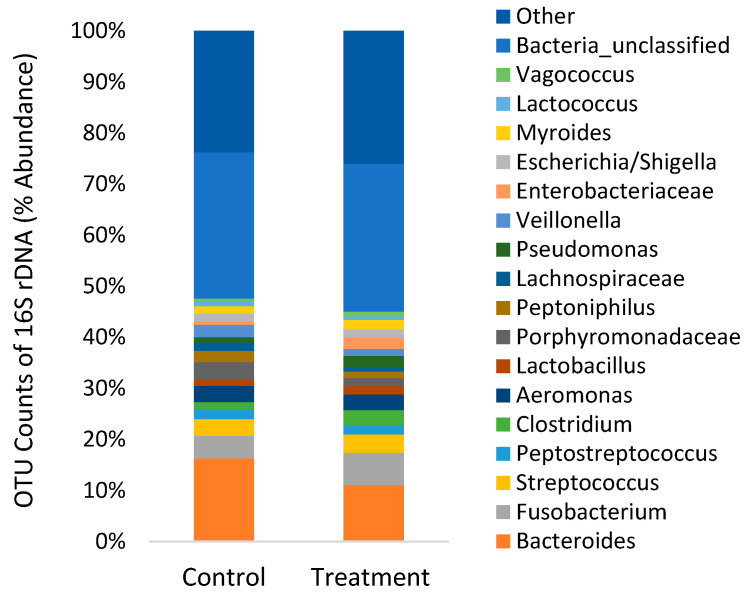
Relative abundance of OTUs on the genus level clustered from 16S rDNA from the gut digesta (*n* = 5) of rainbow trout.

**Table 1 microorganisms-09-02063-t001:** Ingredient and proximate chemical compositions of the test diets.

Ingredient	g/kg
Fish meal-anchovy, 67% CP	50.0
Poultry by-product meal, 60% CP	150.0
Distillers dried grain soluble, 23% CP	100.0
Corn gluten meal, 60% CP	50.0
Dehulled soybean meal, 48% CP	255.0
Corn	125.0
Wheat, grain	125.0
Wheat middling	80.0
Fish oil *	20.0
Soybean oil	20.0
Premix	10.0
Lysine HCl	4.3
Methionine	1.7
Choline chloride	3.0
Ca(H_2_PO_4_)_2_	5.0
NaCl	1.0
**Proximate composition**	
Dry matter, %	91.3
Crude protein, %	37.4
Lipid *, %	9.0
Ash, %	6.8
Gross energy, MJ/kg	16.2

* An additional 100 g/kg (10%) of fish oil was added to meet requirements for rainbow trout.

**Table 2 microorganisms-09-02063-t002:** Histology of the proximal and distal intestine of rainbow trout (*n* = 9) after feeding the control and treatment diets for four weeks.

	Proximal		Distal			*p*-Value ^1^	
	Control	Treatment	Control	Treatment	SE	Proximal	Distal
Villi length (µm)	645	686	989	1039	193	0.035	0.278
Villi width (µm)	155	158	150	149	10	0.497	0.968
Edema	0.67	0.70	0.22	0.00	0.59	1.000	0.343
Inflammation serosa	0.00	0.00	0.00	0.00	0.00	1.000	1.000
Inflammation submucosa	0.20	0.10	0.00	0.00	0.27	0.583	1.000
Inflammation lamina propria	0.20	0.10	0.22	0.00	0.41	0.583	0.343
Vacuolization	1.00	1.00	2.78	3.00	1.01	1.000	0.343
Goblet cells	2.30	2.80	1.22	1.00	0.93	0.139	0.343
Mitoses	1.50	1.10	0.22	0.20	0.81	0.239	0.954
Necrosis/apoptosis	0.30	0.10	0.11	0.00	0.34	0.301	0.343

^1^ *p*-values from either a *t*-test for normally distributed data or Wilcoxon test for non-normal data.

**Table 3 microorganisms-09-02063-t003:** Alpha diversity of Operational Taxonomic Units (OTUs) clustered from 16S rDNA of bacteria from gut digesta of rainbow trout.

	Control	Treatment	SE	*p*-Values
Good’s coverage	0.989	0.989	0.001	0.841
No. of taxa	483	448	104	0.691
Shannon diversity	3.95	3.91	0.05	0.781
Chao-1 richness	1833	2166	91	0.210

**Table 4 microorganisms-09-02063-t004:** Linear discriminant analysis effect size (Lefse) of indicator bacteria species that were significantly (*p* < 0.05) associated with each group.

Diet	Phyla	Family/Genus	LDA	*p*-Value
Control	Bacteroidetes	*Bacteroides*	4.142	0.020
	Bacteroidetes	Porphyromonadaceae	4.144	0.009
	Firmicutes	*Sporosarcina*	3.246	0.048
	Firmicutes	*Veillonella*	4.046	0.018
	Proteobacteria	*Aeromonas*	3.867	0.041
	Proteobacteria	*Acinetobacter*	3.056	0.050
Treatment	Firmicutes	*Streptococcus*	3.700	0.041
	Fusobacteria	*Fusobacterium*	4.175	0.040
	Proteobacteria	*Escherichia/Shigella*	3.959	0.045

LDA; linear discriminant analysis.

## Data Availability

Data is available upon reasonable request. Microbiome data is publically available on the NCBI Sequence Read Archive (http://www.ncbi.nlm.nih.gov/bioproject/767341, accessed on 29 September 2021).

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
