# Peer review of "Dietary Microencapsulated Blend of Organic Acids and Plant Essential Oils Affects Intestinal Morphology and Microbiome of Rainbow Trout (Oncorhynchus mykiss)"

_microorganisms, 2021, doi:10.3390/microorganisms9102063_

Round 1

Reviewer 1 Report

Microorganisms

Manuscript: Microorganisms-1394086

The authors studied the intestinal morphology and microbiome of rainbow trout that fed a diet containing 300 mg/kg of a microencapsulated blend of organic acids and essential oils. Proximal intestinal villi length was significantly increased. Next generation sequencing of the 16S rDNA showed no differences in alpha and beta diversity. The increase in villi length and reduction of specific pathogens indicates that feeding a microencapsulated blend of organic acids and essential oils improves gut health. Reduction in Aeromonas hydrophila and Acinectobacter indicates this treatment may be effective at reducing opportunity pathogens in the gut of rainbow trout and may serve as part of an effective strategy to reduce antibiotic use in aquaculture. This manuscript provides novel information in the field.

  1. Page 3, Line 99: Check the term and unit “total suspended solids of 0.4 ±2 mg/L”. It is total dissolved solids or suspended solids?
  2. References: Italicize scientific names.
  3. Page 13, Line 445: Write the name of journal, volume and page number.

Page 17, Lines 580: Write name of journal, volume and page number

Author Response

We thank the reviewer for taking the time to review our manuscript and appreciate the feedback and revision suggestions that has improved the quality of this manuscript. Below are our responses to their comments:

  1. Page 3, Line 99: The term and units are correct. It is a flow-through system, so it has very little total suspended solids from the tank effluent. 
  2. References: grammar corrected.
  3. Page 13, Line 445: details added.
  4. Page 17, Lines 580: details added.

Reviewer 2 Report

Review of 'Dietary microencapsulated blend of organic acids and plant essential oils affects intestinal morphology and microbiome of rainbow trout (Oncorhynchus mykiss)' by David Huyben, Marcia Chiasson, John S. Lumsden, Phuc H. Pham and M. A. Kabir Chowdhury.

The authors provided an excellent experimental study with well thought out laboratory procedures. In this study, the authors studied the gut microbiota and intestinal morphology in rainbow trout, a valuable commercial cultured species in Canada and worldwide. The authors found that that feeding a microencapsulated blend of organic acids and essential oils can improve gut health of Oncorhynchus mykiss expressed as proximal intestinal villi length. These results are of great importance for rainbow trout aquaculture.

Pg 1 Ln 11. As I can see from weight data, the authors used young fish. This should be mentioned in the text and Abstract.

Pg 1 Ln 16: Suggest changing ' Next generation' to ' Next-generation'

The authors should format their citations and references according to Instructions for Authors.

Pg 3 Ln 117: The authors should provide length data for the fish.

Pg 4 Ln 122: Suggest changing ' was collected' to ' were collected'

Pg 6, Fig. 1. Red font is too small. The authors should increase the font size.

Pg 9 Ln 281: The authors should provide standard error for this value "97.6".

Pg 9 Ln 284-293: This section is redundant and should be deleted.

Pg 10 Ln 343: Italicize 'In vitro'

Pg 12 Ln 418, 422, 425, 435: Italicize scientific names

Pg 13 Ln 451, 452, 460, 465, 468: Italicize scientific names

Pg 14 Ln 492, 495: Italicize scientific names

Pg 15 Ln 509, 510, 523: Italicize scientific names

Pg 16 Ln 538, 541, 546, 552, 553: Italicize scientific names

Pg 17 Ln 568, 577, 580, 587, 588: Italicize scientific names

Author Response

We thank the reviewer for taking the time to review our manuscript and appreciate the feedback and revision suggestions that has improved the quality of this manuscript. Below are our responses to their comments:

Pg 1 L 11: "juvenile" has been added to the abstract and text to indicate these are young fish.

Pg 1 Ln 16: grammar corrected.

Pg 3 Ln 117: fish length data was added.

Pg 4 Ln 122: grammar corrected.

Pg 6, Fig. 1. Figure 1 font size was increased.

Pg 9 Ln 281: We used "pooled SE" so the SE applies to both values. We included this in the text to be more clear.

Pg 9 Ln 284-293: This section was deleted.

Pg 10 Ln 343: Italicize 'In vitro'

Pg 12 Ln 418, 422, 425, 435: Italicize scientific names

Pg 13 Ln 451, 452, 460, 465, 468: Italicize scientific names

Pg 14 Ln 492, 495: Italicize scientific names

Pg 15 Ln 509, 510, 523: Italicize scientific names

Pg 16 Ln 538, 541, 546, 552, 553: Italicize scientific names

Pg 17 Ln 568, 577, 580, 587, 588: Italicize scientific names